# The Bondo Society as a Political Tool: Examining Cultural Expertise in Sierra Leone from 1961 to 2018

**Aisha Fofana Ibrahim**

Institute for Gender Research and Documentation (INGRADOC), Fourah Bay College, University of Sierra Leone, 00232 Freetown, Sierra Leone; mamaisha@gmail.com

**Abstract:** This paper focuses on the politics of the Bondo—the competition among social groups for an exclusive influence on the National strategy for the reduction of female genital mutilation/cutting (FGM/C). In the first part, this paper shows how the Bondo—a women's only secret society—has become a site of contestation for not only pro- and anti-FGM/C advocates, but also elite male politicians who have, since independence in 1961, continued to use the Bondo space for political gains. The use of the Bondo for political leverage and influence pre-dates independence and is as old as the society itself. The second part of this paper discusses the legitimacy of expertise as central to this debate, in which each group competes to become the leading expert. Thus, even though human rights/choice discourse currently dominates the FGM/C debate, traditional expertise remains valid in the formulation of community by-laws as well as state policies and laws. This can be seen in the recent attempt by the state to develop a National Policy for the Reduction of FGM/C in which the expertise of all three groups was sought. Using data from existing literature and personal interviews, this paper interrogates this contention by describing how the role of cultural experts—especially the Soweis—has been politicized in the stalemate over the enactment of the National Policy for the Reduction of FGC. This paper concludes with considerations about the complexity of Bondo expertise, in which opposing parties use similar arguments to evoke the human rights discourses on women's rights and bodily integrity/autonomy. It argues that a better knowledge of these dynamics as they develop in Sierra Leone and other African countries would be useful to the European jurisdiction.

**Keywords:** Bondo; FGM/C; National Strategy; cultural expertise; human rights

## 1. Introduction

In Sierra Leone, secret societies not only offer gendered and cultural spaces but—more importantly—operate as a crucial site of political power, making profound decisions about community wellbeing such as the promulgation of laws and how initiations are carried out (Fanthrope 2006; Pemunta and Tabenyang 2017). The trajectories and nuances of the Bondo—a women's only secret society that practices FGC—can be best interrogated through an understanding of the political economy of the practice. This is because the Bondo is both an economic and political enterprise, with Bondo leaders working hand in hand with traditional male authorities in different political spaces. Bondo leaders are often self-seeking and maximize their utility by participating in political activity for both political and economic gains, as manifested in the role they play in the preparation of girls for marriage to powerful political leaders. Dating back hundreds of years, the society has always been dependent on the male patronage of paramount chiefs and politicians, the majority of whom are members of male secret societies. Male power and control is manifested through the society's symbiotic relationship with the all-male Poro secret society, as well as the Bondo society's location within traditional power structures. At the economic level, the Bondo is arguably an economic enterprise with heads earning their livelihood through fees charged for initiations and other related fines. As Bosire (2012) argues

"the debates about FGC are not simply about" "the defence of culture," "they are also about livelihoods" (p. 90).

Unlike in other places, in Sierra Leone, excision takes place within the context of a secret society—the Bondo Society. Excised women and girls automatically become members of the Bondo, which is operated by "powerful" women called 'digba' or 'Sowei' who have consistently laid claim to cultural expertise with regard to the practice. The membership aspect suggests that the society transcends the act of cutting, but yet this cutting is an important aspect of its rituals. According to Bosire (2012), the Bondo is a "repository of gendered knowledge that bequeaths members with privileges and power safeguarded by secrecy" (p. 51). The oath of secrecy used to be so strong that initiates were afraid to openly discuss the procedure—a taboo that no longer prevails. Moreover, as Bosire (2012) further elaborates, "soweis reported that Bondo initiation increasingly involved only the ritual cut and customary Bondo teachings were limited. The commercialization of Bondo initiation has also, in one aspect, had the effect of whittling down the traditional symbolic authority accorded to soweis" (p. 88). There are two main procedures performed in Sierra Leone—sunna, the removal of the hood of the clitoris, with the body of the clitoris remaining intact; and clitoridectomy or excision, which is the removal of the clitoris and all or part of the labia. Bjalkander et al. (2013), from a study that involved genital examinations of women who had undergone the procedure, posit that even though the majority of respondents may have wrongly described the form of cutting experienced, forms of type 1 and 2 continue to be the most prevalent forms of FGM in Sierra Leone. This is to say that other types do exist, but are very rare. Depending on the region and person asked, the procedure is justified on the basis of tradition, religion, rites of passage, health and assurance of virginity before marriage as well as marital fidelity, but with the majority arguing that it is a tradition and culture dealing with rites of passage that should be upheld.

## 2. The Bondo Society of Sierra Leone

The West African region is home to a number of all-male and all-female secret societies, which continue to play an important role in communal life especially in the rural areas. As Mgbako et al. (2010) explain, "Secret societies have fulfilled a number of philosophical, economic, political, social, religious, and educational functions in their communities throughout history, and they continue to play a significant role in contemporary West Africa" (p. 118). The Sande—traditionally an all-female society—predominates in the Mano River region (Sierra Leone, Liberia and Guinea) and is shrouded in secrecy and female power. Although genital cutting is not uniform across these countries and among ethnic groups, it remains a pre-requisite for membership of the secret societies. Commonly referred to as the Bondo Society in Sierra Leone, initiates of the Bondo take an oath of secrecy and are not allowed to disclose what goes on in the Bondo bush. The consequences of breaking the oath are believed to include an individual and/or family curse and infertility. Non-members are neither allowed into the Bondo bush nor allowed to discuss Bondo affairs or offend a Bondo member.

The Bondo bush—the sacred place for society women—is often located in the forest far away from daily life. This dynamic has changed considerably, with Bondo bushes now springing up in the middle of towns and behind people's backyards. The Bondo is very hierarchical and has a well-defined organizational structure, with a head Sowei and other members in various other positions. Soweis are expected to undergo an intensive 2 to 3-year period of training in how to perform genital cutting and apply medicinal remedies. Soweiship is often hereditary and handed down from generation to generation. For Ahmadu (Sulkin 2009) "the institution itself is synonymous with women's power, their political, economic, reproductive and ritual spheres of influence. Excision, or removal of the external clitoral glans and labia minora, in initiation is a symbolic representation of matriarchal power" (p. 14).

However, the Bondo society has had its own fair share of problems in a modernizing society in which people are progressively accessing information from within and without. On the one hand, it is an institution that contributes to social cohesion in communities by maintaining law and order

and, on the other, it is a site of struggle. As Bosire (2012) points out, "the Bondo, like all cultural phenomena, is also a site of social struggle. The acquisition of leadership positions within the Bondo is a process in which some of the inherent tensions and debates and cultural uses of the Bondo, the meanings associated with some Bondo values, and the changing structures of the Bondo are played out. The tensions in the Bondo are partly engendered by the changes that the Bondo is undergoing in contemporary post-war Sierra Leone and in the face of FGC eradication discourse" (pp. 76–77).

The Bondo is embodied in a patriarchal ideology that associates women with the body, and with blood and flesh. As Ahmadu (Sulkin 2009) postulates, "Bondo women elders believe and teach that excision improves sexual pleasure by emphasizing orgasms reached through stimulation of the g-spot, which is said to be more intense and satisfying for an experienced woman. Excision of the protruding clitoris is said to aesthetically and physiologically enhance the appearance of the vulva and facilitate male/female coitus by removing any barrier to complete, full and deep penetration" (p. 16). Over time, women's bodies have been despised, tolerated, and exalted, depending upon the immediate context and prevailing politics. Male desires and politics have been scripted onto the female body and women have unwittingly accepted these scripts (Lee and Sasser-Coen 1996, p. 6). Arguably, the Bondo society of Sierra Leone—in which women believe they yield a particular power, (for they alone perform excisions)—not only perpetuates the use and misuse of the female body, but also male desires and politics.

The Bondo symbolizes a girl's entrance into female fecundity and adult female sexuality and serves as a social marker of movement from being a girl to a woman. The question that has always marked the debate is—does a young woman have to lose a vital organ to make such a move? Moreover, what does it mean to experience such a crucial signifier of womanhood in a society that devalues women, especially when this devaluation occurs through cultural scripts associated with the body? (Lee and Sasser-Coen 1996, p. 5).

This paper interrogates the politics of the Bondo at both the local and national levels, the activism around the Bondo by members of the society from two different camps—each of which claims cultural expertise on the society and the politics around the formulation of the National Strategy on the Reduction of FGM/C—and how these are interconnected. Central to this debate is the question of how cultural expertise is produced regarding Bondo society—what the Bondo means and how it is experienced in contemporary Sierra Leone. Like every culture, the Bondo's has changed over the years, moving from a one-year experience where matured girls are taught the art of traditional healing, home making, motherhood and sex education, to—in many cases—a one week experience where only excision takes place. The latter cases consequently trigger the highly political questions—if all that is left of Bondo culture is excision, how is it a culture worth maintaining? Why cannot the Bondo be revived with all its secret powers, but without the harmful practice of cutting? Moreover, in a globalized world where cultures are no longer clearly distinct—yet in which migrants decide to adhere to cultural practices that are criminalized in their host countries—the role of the cultural expert becomes increasingly important and vital for the administration of justice. It is therefore important that cultural expertise go beyond understanding the nuances of specific cultural practices, but also include an interrogation of the expert's political/ideological stance on the issue. In Sierra Leone where the issue of FGC is highly politicized, there seems to be no middle ground between pro- and anti-FGC campaigners and this certainly problematizes cultural expertise.

## 3. Methodology

Data for this paper was collected between 2016 and 2018 through interviews conducted with 10 members of the Forum Against Harmful Practice—a coalition of national and International organizations working towards ending female genital mutilation (FGM), reading reports, newspaper and academic articles and taking notes at various workshops and meetings held on the National Strategy on the Reduction of FGM/C. I tracked public discourses about FGC/M from both sides of the debate on TV, radio and other media sources. These included one-hour programs in which Soweis

and members of the Sowei council were given the platform to explain their position—those with only anti-FGM/C campaigners, mixed panels and those with members of the public. These interviews were a common occurrence in the media and they opened up the conversation around FGM/C. FAHP members were interviewed to fully understand the nature and scope of their campaign, how they engaged with Bondo leaders and traditional authority and their role in the development of the Strategy.

## 4. The Politics of the Bondo

The use of the Bondo for political leverage and influence pre-dates independence and is as old as the society itself. Madam Yoko—paramount chief of the Kpa Mendes and the quintessential cultural expert on the Bondo—used the society to gain political power and influence during her reign in the 1800s. MacCormack (1974) posits that—as a Sowei and paramount chief—she used the society to consolidate her rule by pairing off beautiful Bondo initiates with powerful men. Existing within a hostile patriarchal environment, she bargained with patriarchy in many ways, to not only stay in power, but to become so powerful as to be considered almost an equal by other male rulers. What Kandiyoti (1988) refers to as "patriarchal bargaining" still forms the core of the Bondo women's access to power and influence. If they are not siding with paramount and local chiefs in the oppression of community members through fines and sanctions, they are selling the notions of solidarity and sodality among women of the society and thus presenting themselves as a formidable constituent to politicians vying for office. Since independence, politicians have used the strategy of financially supporting the mass initiation of young girls into the Bondo society, just before an election. They believe that this practice guarantees them large numbers of votes from community members whose financial burden is reduced through this act of "kindness." Initiating one's children into the Bondo society is often financially traumatic for many poor families who often end up in debt, as the Bondo can be a very expensive investment that arguably bears no fruitful gains. Initiators, chiefs and community leaders, as well as initiates, have to be lavished with food and gifts and cash payments have to be made to initiators and fees paid to paramount chiefs, who have to approve any ceremony in the community. Politicians willingly cover all of these costs, because they understand that in a society embedded in the traditional belief system of "akeh"—in which people believe that there are dire consequences for going back on one's word or betraying the trust of a person—or the belief of "one good turn deserves another," very few will dare to provoke what they consider to be the wrath of the gods by reneging on their promise to vote for these politicians. Interestingly, many of these politicians will pay to have other people's children excised, while they keep their own children out of the society and are still not held accountable by Soweis.

The fear of losing the rural vote has resulted in many politicians flirting with both sides of the argument. They may be against the FGM/C side of the Bondo, but publicly parrot the "respect for culture" mantra. This has in many ways hindered the promulgation of any progressive legislation on the practice and continues to pit women against each other. For example, the Child Rights bill of 2007 clearly specified a ban on FGM/C, but politicians were split on whether the FGM/C clause should be removed before it is enacted. In the end, the act was introduced, but without the FGM/C clause.

Since independence to date, not only has the political class—including women—failed to openly condemn the cutting aspect of the Bondo, but it has tacitly supported Bondo—not only by paying for mass initiation ceremonies, but by appointing/electing them to political positions such as community chairladies and party agents. Only in rare cases will a politician speak against the Bondo and that only happens when they know that they have an overwhelming influence over their constituents. At an anti-FGM/C event in 2015, the then minister of works—Kemoh Sesay—publicly condemned FGM/C and vowed to support the campaign for the abandonment of the practice. He also claimed that 70% of politicians are against the practice, do not excise their daughters and are "playing politics" with people's lives, because they believe that publicly condemning the practice is tantamount to political suicide.

Soweis have been able to capitalize on this undue fear of politicians to build a support base to counter anti-FGC discourse and advocacy[1]. They understand that the majority of proponents are uneducated and live in rural communities where these celebratory rites bring respite and joy to their lives of everyday drudgery. Soweis politicize the anti-FGM/C campaign, claiming first and foremost that—as practitioners—they have the sole rights to cultural expertise and further that those against cutting are stooges of the west, who no longer have any respect for "African ways" of being. As far as they are concerned, the questioning of the Bondo and its practices by lay persons and the encouragement of "non-natives/initiates" to discuss the practice is, in itself, abominable. The Bondo is not an issue discussed among initiates/members, let alone "airing" and sharing detailed information about the practice with "outsiders" and non-initiates. The evocations of race and location in this debate have provoked a number of political outbursts, which are typified by the recent impasse between anti-FGM campaigners and the most recent Minister of Gender Affairs on the validation of the draft Strategy on the Reduction of FGM/C. The minister—ironically a non-initiate—aligned herself with the Soweis, claiming that the anti-FGM campaign is a form of western propaganda which was not locally initiated, and which mainly exists because advocacy groups want to tap into funds being provided by western countries.

At the local level, the symbiotic relationships between the Bondo and the Poro and the Bondo and the chiefs are both political and economic. The paramount Chief is the only person who can authorize the construction of Bondo bushes or the initiation of girls. Bondo licenses, fees and fines for the infringement of the society's rules, form a huge chunk of revenue for Paramount chiefs. In addition, Soweis always have a coterie of young girls from whom chiefs can choose to become one of their many wives. As a part of traditional structures, Soweis often advise chiefs on issues affecting women in the community, broker peace and act as mediators between erring persons and the chief, thus building their social and political capital in the community.

Bondo leaders do play an important role in the exclusively male institution of the Poro, as the integral office of the "Mabole" is traditionally occupied by senior Bondo leaders, who are believed to have knowledge of the traditional healing herbs needed in the performance of Poro rituals. The reverence of the Bondo to the Poro society is articulated in a very popular saying in Sierra Leone—"Nar Bondo born Poro" (Bondo gave birth to Poro). However, this symbolic power is subordinated when a woman becomes a paramount chief and is expected to give up her Bondo membership to be initiated into the Poro, thereby clearly manifesting that leadership is masculine and male-dominated. Moreover, it has been argued that although collaboration between the Poro and the Bondo offers women some form of power, they are actually serving the interests of the patriarchal order by ensuring social cohesion through the formation of docile bodies (Phillips 1995; Pemunta and Tabenyang 2017; Fanthrope 2006). Through the practices of the Bondo, women's and girls' bodies and minds are trained and molded in the context of prevailing systems of power, presenting them with important lessons about feminine bodies, a woman's place, and desire. Women internalize these discourses and reproduce them, thereby maintaining oppressive, gendered social relations.

There was a great deal of hope that there would have been some major reforms to the Bondo society following the Ebola outbreak in 2015. This is because during the Ebola crisis, the paramount chiefs and some councils of Soweis—the custodians and experts of the culture—enacted by-laws that banned or postponed all forms of initiation ceremonies, with the dire consequences of fines and arrests if violated. This ban was reasonably effective because of the severity of the disease and the vigilance of the chiefs and Soweis. In addition, the president made a very hopeful statement during his speech on 7 November 2015 while declaring the end of Ebola, in which he stated that "a new beginning warrants

---

[1] I say undue fear because Rugiatu Neneh Turay—a very vocal anti FGM/C campaigner—12 years ago contested and won a council seat in Port Loko, where her organization the Amazonian Women's Initiative is located. Moreover, Sierra Leoneans—especially those in the rural areas—often vote in patterns whereby political party and ethnicity supersede every other consideration.

that traditional practices that have a negative impact on health, and which were discontinued during the outbreak, should not be returned to." Even though the president did not specifically name FGM/C, it was assumed by anti-FGM/C campaigners that FGM/C was one of such traditional practices to which he was referring. In as much as this statement was used as a campaign tool, there continues to be no political will to ban FGM/C—thus leaving the fight mainly between society sisters and non-cut advocates in the anti-FGM/C camp. In addition, two Sierra Leonean women scholars from an ethnic group that does not practice FGM/C—the Krios—have also contributed greatly to the understanding of the practice through research and advocacy.

## 5. When Society Women "Fight" Claiming Cultural Expertise and Human Rights

The debate around the merits and demerits of the Bondo (especially the FGC aspect of it) by pro- and anti-FGM/C activists in Sierra Leone is interestingly taking place mostly among members of the society. This is no surprise because, according to UNICEF (2016), 90% of Sierra Leonean women and girls have been excised and are thus members of the Bondo Society. Because women leading the campaign for the abandonment of the practice are members of the society, the insider/outsider labeling is difficult to make, hence the recourse to the branding of anti-FGM/C advocates as sell-outs to an imperialist agenda who limit the term "harmful traditional practice" to the excision of women whilst being mute on male circumcision. As far as they are concerned, the campaign against harmful traditional practices in Sierra Leone should not be limited to the Bondo society, but also extended to the Poro society.

Globally, the practice has generated heated debates between cultural relativists and Universalists. According to Renteln (2004), the doctrine of cultural relativism holds that there are no value judgments that are objectively falsifiable independent of specific cultures, and as such, moral judgments and social institutions in any one society are exempt from legitimate criticism by outsiders. On the other hand, the Universalist stance holds that certain individual rights are so fundamental to humankind that they should be upheld as universal rights whose breach is subject to condemnation and—in certain instances—punishment through legislative force (p. 127). Relativists thus see the practice of FGM/C from the vantage point of a ritual that signifies an important event in individual and group life. Whereas to many, like me, who subscribe to the Universalist position, the practice is viewed as an act of violence and violation of the human rights of women.

Although I identify with the Universalist view on the issue, I still question its sensationalization by international women's forums and agree with Toubia (1988)—a Sudanese scholar—who argues that these feminists have acted "as though they have suddenly discovered a dangerous epidemic, which they then sensationalized, in effect creating a backlash of over-sensitivity in the concerned communities. They have portrayed it as irrefutable evidence of barbarism and vulgarity of undeveloped countries. [and] it became a conclusive validation of the view of the primitiveness of Arabs, Muslims, and Africans all in one blow" (p. 101).

Cultural expertise are claimed by both sides of the divide, either because they have undergone the procedure, experienced the rituals or because they are the initiators and heads of the society. As cultural experts on the society, both sides of the campaign respect and believe in the importance and necessity of the Bondo society in the cultural identity of Sierra Leonean women. Where they differ is that anti-FGM/C activists want the society to abandon cutting, whereas the pro-FGM/C proponents believe that cutting is intrinsically intertwined with the Bondo and therefore find it difficult to imagine the Bondo without cutting. Both groups utilize human rights discourse, forgetting the fact that rights "including so-called universal ones, are not natural and eternal but always emergent and historically specific" and that rights always need to be contextualized, interpreted and negotiated (Cowan et al. 2001). Post-war reconstruction created a space for human rights to take hold in Sierra Leone and has offered an overarching, normative framework for civil society organizations to advocate for various rights. Thus, it is not surprising that there are claims of human rights violations from each side of the divide. The anti-FGM/C camp highlights the forceful initiation of children as a gross

violation of the rights of children and evokes all the international and national treaties that protect the bodily integrity of women and children Sierra Leone has ratified, while the pro-FGM camp claims that the anti-FGM/C camp—who they have labelled the Zero Tolerance Propaganda Campaign—are violating their rights to cultural autonomy. Dr. Fuambai Sia Ahmadu—a Sierra Leonean born medical anthropologist and vocal Pro-FGM/C campaigner, who is the founder and intellectual guru of the Sierra Leone Women are Free to Choose (SLWAFC) movement, and who as recently as 6 February 2018 (International Day of Zero Tolerance on FGM) launched its first Female Circumcision Awareness Week in Sierra Leone in collaboration with the Sowei council—defends the pro-FGM/C campaign as such:

> I have been at the forefront of global debates and activism to counter the harmfulness and hypocrisy of FGM campaigns and to help restore the rights, autonomy and dignity of women who support or choose to uphold female circumcision as a religious or cultural practice. In Sierra Leone, this led to the formation of Sierra Leone Women are Free to Choose, to protect the fundamental constitutional and human rights of sowies as well as Bondo women who continue to uphold female circumcision as an important expression of gender identity or womanhood. (Interview 7 February 2018)

Anti-FGM/C campaign groups in Sierra Leone—such as the Forum against Harmful Practices (FAHP), which is a coalition of national and international non-governmental organizations working on the abandonment of FGM/C—have tried to incorporate a wide range of cultural custodians and experts including religious leaders, political leaders, medical professionals, paramount and section chiefs and the Soweis themselves in this campaign. They have placed FGM within a broader social justice agenda in which the government plays a central role in the protection of its citizens and have consistently used the Child Rights Act (2007)—which stipulates that the age of consent is 18 and criminalizes underage initiations—to hold the government and Soweis accountable for the initiation of underaged children. The campaign against the forceful initiation of children is very good, but falls short of addressing societal ostracism as the question remains—if a woman or girl's social identity is tied to the practice and she is ostracized for not participating, is that not also a form of force? The inability or unwillingness of the government—through the police and justice system—to enforce this aspect of the act shows how deeply engrained Sierra Leonenan society is in the belief systems of the Bondo society. It is common knowledge that ideologies and belief systems are most effective when most taken for granted. They resist correction and critique by making the status quo appear natural, "the way things are", rather than the result of human intervention and practice.

Anti-FGC campaigners also argue that the procedure is "a part of a continuum of patriarchal repression of female sexuality, which has been repressed in a variety of ways in all parts of the world throughout history and up to the present time" (Dorkenoo 1994, p. 29). They posit that when pro-campaigners argue that FGM is a culture-specific practice and a traditional practice that should be maintained, they fail to understand the might of the patriarchy and the status of FGM/C in this continuum of the patriarchal repression of female sexuality. They go on to point out the connections between the patriarchy and a string of repressive practices over time: the locking of the labials of female slaves with rings in ancient Rome; the chastity belts introduced by the Crusaders during the twelfth century; and, as late as the 1950s, the surgical removal of the clitoris as a "cure" for various ailments such as insomnia, sterility, lesbianism, masturbation and other supposed sexual deviances. They assert that the thousands of cosmetic surgeries performed today in Western societies indicate that the commodification, objectification and control of women are far from over. They cite the patriarchy as essential to the reasons why the bodies of women from different parts of the world, and from different religious, economic and educational backgrounds are "made" and expected to conform to societal expectations. In essence, they believe that sexualization in patriarchal societies involves a loss of female power, autonomy, and efficiency, and an imposition of norms and restrictions that are internalized by both men and women (Lee and Sasser-Coen 1996, p. 103).

Anti-FGM advocates tend to emphasize the health risks of the procedure, drawing their expertise from experience and research, both of which demonstrate that excised women and girls face many

complications including urinary retention, bleeding, pain, septicemia and vaginal fistula, which in most instances lead to social ostracism and divorce. They argue that the procedure is mainly performed on children as young as one—who actually have limited voices or power—who are forced to undergo a procedure that can potentially leave them physically and emotionally traumatized. They see women's bodies as sites of struggle that involve both compliance and resistance to normalizing discourses and understand that the control of women's bodies in both public and private spaces is essential for the maintenance of patriarchal societies.

As convincing as the arguments raised by anti-FGC campaigners are, those who vehemently fight against its abolition are women and not men. It will therefore be simplistic to present a picture of the African woman as wholly subservient, passive, and "voiceless"; one whose sexual and reproductive potential is controlled by men and whose genitals are mutilated in silence without protest. Thus, because women have the upper hand in determining when, how, and where a girl will be excised, it is often difficult to make people understand that the practice is based on a patriarchal value system. In Sierra Leone, Bondo women yield power not only because they alone perform these procedures, but also because they have been able to bargain with the patriarchy in varied ways. Abusharaf (2000) posits that in the Sudan, the ritual becomes an important affirmation of one generation of women's authority over another. She cautions that this should not be dismissed as an expression of false consciousness, in which women perpetrate their own subjugation, nor can the motive behind FGM/C be traced to a single patriarchal value.

Pro-FGM/C campaigners see the procedure as an important marker in the transition to adult femininity, purity, and marriageability and—even though this is far from the reality—see the Bondo bush as a place where initiates are taught the art of home keeping, good social relations with in-laws, sex education, child-bearing and aspects of motherhood and traditional medicine. The organizing of initiators into a structured body called the Sowei council—formed in 1993 and headquartered in Freetown, but with branches in all the districts—gave this group a formal public presence that can insert itself to counter on-going anti-FGC rhetoric and actions. Since its formation, Soweis have been formally invited to conferences and workshops, and have been able to engage the media to advocate on their own behalf. The more visibility they gained, the more defiant they have become, and the more the political wind blew in their favor, the more emboldened they have become. What seems to aggravate the Soweis the most is that the "secret" of the Bondo is now in the public sphere (media, meetings, workshops, etc.) and is discussed by non-initiates and non-experts. This they see as an abuse of their culture and a violation of their right to cultural integrity.

Irrespective of the WHO's definition that—by medical standards—the removal of a healthy normal organ from a human body when there is no medical or aesthetic reason is a mutilation, pro-FGC campaigners of Bondo in Sierra Leone take offence to the word mutilation and consider it a racist and degrading epithet for an experience they find somehow empowering. They argue that the term "mutilation" does not necessarily differentiate between the forms and degree of severity practiced by different societies. As Ahmadu (2018) argues:

> The fact is that the majority of circumcised women support female circumcision just like the majority of circumcised men support male circumcision. For most of the population of women in Sierra Leone, the term FGM is a huge affront to our identity and an unacceptable insult against us, our mothers and female elders. (Interview 7 February 2018)

The pro-campaigners further argue that the use of "choice" by western anti-FGM campaigners is very selective, especially in relation to cosmetic surgery and FGC. They point out that there is an overwhelming silence on acts of "mutilation" going on in the western world through cosmetic surgeries such as labia reduction and other forms of "designer vagina" surgeries. Many of these surgeries, like FGC, they argue, are performed on underaged girls for whom parental consent is deemed enough before surgeries take place. Thus, the question they ask is—why is parental consent by parents of young girls who undergo FGC not enough? Dr. Fuambai Sia Ahmadu (2016) in another interview had this to say on the issue:

> While our African governments are busy succumbing to pressure from western women to outlaw our traditional female genital aesthetic practices, western countries have developed a flourishing female genital cosmetic surgery industry, using our own operations as the aesthetic standard.
>
> And, instead of fighting to defend the rights of our mothers and grandmothers, many of us who are western educated have given carte blanche for them to be stripped, degraded and punished by and for the sake of the very white women whose own mothers and daughters are now freely opting for the same procedures.
>
> I have always said, Sierra Leone is the ground zero where modern western feminism meets the power of ancient Bondo society. As you can see, I've placed my bets on Bondo. (Interview 8 February 2016)

The racial undertones and bias of the anti-FGM/C campaign have also been highlighted by many scholars. For example, Ehrenreich and Barr (2005) argue that

> " ... the mainstream anti-FGC position is premised upon an orientalizing construction of FGC societies as primitive, patriarchal, and barbaric, and of female circumcision as a harmful, unnecessary cultural practice based on patriarchal gender norms and ritualistic beliefs. ... Lambasting African societies and practices (while failing to critique similar practices in the United States) ... essentially implies that North American understandings of the body are "scientific" (i.e., rational, civilized, and based on universally acknowledged expertise), while African understandings are "cultural" (i.e., superstitious, un-civilized, and based on false, socially constructed beliefs). [Yet] neither of these depictions is accurate. North American medicine is not free of cultural influence, and FGC practices are not bound by culture—at least not in the uniform way imagined by opponents". (Cited in Earp 2014)

**6. The National Strategy for the Reduction of FGM/C: A Site of Contestation**

In order to fulfill its obligations as a signatory to the Convention on the Elimination of Discrimination against Women (CEDAW), the Convention of the Rights of the Child (CRC) and the Maputo Protocol, the government—with the support of UNICEF, through its Minister of Social Welfare Gender and Children's Affairs (MSWGCA)—in 2014 commissioned the development of a National Strategy for the Reduction of FGM/C (2016–2020), with three key pillars: Pillar 1—Creating an Enabling Environment for FGM/C abandonment; Pillar 2—Strengthening National Capacities to prevent FGM/C and care for those living with FGM/C; and pillar 3—Sustained Community Commitment to FGM/C abandonment. The position of the government is made clear in the introduction to the strategy which states that "The Government of Sierra Leone, whilst upholding the noble values of the Institution of Bondo recognizes the practice of FGM/C, which is presently part of the initiation activities for Bondo membership, as a health burden and a violation of the human rights of children and women" (National Strategy 2016). Recognizing the challenge, the government further states in the foreword that,

> We are aware that postponing FGM/C does not mean reduction. For us to achieve reduction, we need to reduce the incidence of new cases of girls undergoing FGM/C. In this strategy, we consider measures which whilst celebrating and upholding the wholesome aspects of the Bondo Institution seek to remove the harm from the initiation ceremony and create alternative new spaces where adolescent girls can be publicly recognised as women and are identified with their ethnic groups. (National Strategy 2016)

Even as sensitive to the wishes of both sides of the divide as the document has tried to be, it has become the most contentious document in the FGM/C debate and is still on the shelves in the Ministry,

yet to be validated and implemented. The politics around the strategy continues to deepen the divide between both camps, with each hoping for a Minister that is sympathetic to its cause.

While the anti-FGM/C camp claims that the process was very inclusive, the pro- camp insist that it was not and that the views of practitioners were not taken into consideration. Reports, however, show that in January 2014, validation workshops were conducted in all 13 administrative districts in the country to ensure the buy-in of all stakeholders—men, women, Soweis, MPs, paramount and other chiefs, religious and political leaders. The aim was for each group to play its own role towards the abandonment of the practice. However, taking into consideration the number of Soweis and Bondo bushes, the argument that many would not have been reached via a single district validation exercise remains valid. For such a contentious issue, continuous engagement is imperative.

The government's recognition of the practice as a violation of international human rights laws and its call for the abandonment of cutting in the rituals of the society are big bones of contention that do not sit well with pro- campaigners. This places responsibility for the practice squarely with the state, which has the duty to ensure that its citizens enjoy full human rights and that means enforcing the law by persecuting violators.

## 7. Conclusions

In the Sierra Leonean context, the "special knowledge" of "so-called 'cultural brokers'" is highly mediated by a subjective position that is politically and ideologically driven. It is almost impossible for such brokers not to take sides if and when called upon to provide expert testimony. The fact remains that the differing views between pro- and anti-FGC campaigners in Sierra Leone fail to recognize that "cultural expertise rests on the ability to distinguish and valorise different cultural forms in a way that resonates with others possessing the same expertise, meaning expert judgements are as much of other people's judgements as of the forms in question." (Kontoyannis and Christos 2010, p. 747). Both camps are so convinced about their positions that, as things stand, no middle ground seems to be feasible, excepting the existence of a strong political will to drive the process. A starting point is a genuine acceptance of the age of consent and the enforcement of the law without political interference. It is apparent that on the level of practice, there remains a diminishing degree of choice for communities and individuals whose traditions have become irrevocably situated in the public arena and—on the level of discourse—silence on the topic no longer seems to be an option, and the choice that remains is between informed and non-informed discussions (Shell-Duncan and Hernlund 2000, p. 3). In Sierra Leone, the debate has permeated every corner and level of society, making it imperative for a critical interrogation of the cultural expertise of both groups and the political and economic dynamics that keep the "fight" going. Why would a group (pro-FGC) so violently defend a society that has over the years abandoned rituals and practices that were believed to make them powerful, but continue to hold on tightly to just one ritual (cutting) that has been described as harmful? And why would another group (anti-FGC) risk their lives, as well as social and political capital to condemn a practice that has been in existence for hundreds of years and which is said to be supported by the majority of Sierra Leoneans? The fact of the matter is that traditional practices are all liable to change—slavery, corsets, and foot binding were once thought to be essential elements of cultures in some parts of the world. Change is already happening in relation to the Bondo, as it is already being threatened by the global anti-female circumcision discourse which is not going to change any time soon. Choice and the age of consent have become central to the debate and that was not the case a few decades ago. The arguments of both sides of the divide are couched in the human rights discourse of choice. The question therefore is how do we situate cultural expertise within the human rights discourse in cases where human rights is evoked to defend traditional practices that have been deemed harmful in non-practicing societies?

**Funding:** EURO-EXPERT-ERC funded project 681814 sponsored the presentation of this essay at the conference entitled "Cultural Expertise in Ancient and Modern History" convened in Oxford by the principal investigator, Livia Holden.

**Conflicts of Interest:** The author declares no conflict of interest.

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
