# Peer review of "The Bondo Society as a Political Tool: Examining Cultural Expertise in Sierra Leone from 1961 to 2018"

_laws, 1961_

Round 1
Reviewer 1 Report
This is certainly an interesting topic, and the author has touched on several of the ethical and political arguments surrounding it. The article details the arguments of pro- and anti- FGM/C constituencies in Sierra Leone. (The author nevertheless stakes a personal claim, essentially taking sides. That can be a tricky move.) The author explains that both sides use human rights discourse, but differently.
Concerns:
There is no real methods section. How does the author come to know all that is asserted in the paper. From newspapers? Interviews?
In the first paragraph, line 31, the author claims that political economy is the best way to study the topic (but doesn’t say why), and then doesn’t really use political economy for the rest of the paper.
I have some concerns that the author over-simplifies the pro-society position. The author says that the societies have lost most of their meaning, aside from the practice of cutting. This is a well-known moderate centrist position in the debate. But this framing, as well as the framing of patriarchal control, both ignore the fact that the women’s society has often been the locus of female political power. It’s no wonder women don’t want to give it up. I also don’t like the conclusion on lines 383 that the debate is one between informed and uninformed positions, especially since the theme of the analysis is expertise.
I really dislike the section around lines 359-363. Its tone is accusatory, almost blaming people for not working to end FGM despite thirteen “validation workshops.” Why assume the efficacy of those workshops? People clearly have good reasons for not abandoning their practices.
Re. assertion on line 172 that the fight is mainly between society initiates, I would add that there is expertise from non-initiates as well. For example, Krio women do not generally join the society, and they frequently use a more Western, legalist, human rights expertise.
Probably most importantly: The central “case” of the paper, the battles over the national strategy for the reduction of FGM/C, take up less than a page! There should be much more detail about the case. One possible solution: perhaps the author could weave together the explanation of the two sides of the issue and the story of the battle over the national strategy. One could be used to further the narration of the other.
If the author wants to continue using expertise as a central theme of the paper, I recommend this review on the topic: Carr, E. S. (2010). "Enactments of Expertise." Annual Review of Anthropology 39(1): 17-32. (Also, I wonder if the author could question her or his own expertise…)
I don’t know the journal at all, but it seems to me this article might not be a good fit. The author talks around lines 22 and 23 about how this work can be “useful to the European jurisdiction” but never explains how in the paper. Indeed, this paper seems to be about gender politics in Sierra Leone more than about either law or expertise.
Smaller points
The author is a universalist around line 190 and a constructivist around lines 204-205.
Kind of odd to only cite Fanthorpe on the political power of women’s societies. Can't the author find someone other than a white man to make that point? And there are much more recent summaries of the debates than Lee and Sasser-Coen, 1996.
Lee and Sasser-Coen is not in the works cited. Is this the article?
Lee, J., & Sasser-Coen, J. (1996). Memories of menarche: Older women remember their first period. Journal of Aging Studies, 10(2), 83-101.
Reviewer 2 Report
This article has the potential to contribute to the debates around FGC, rights and culture and universal concepts versus local lived experiences. The author offers some succinct evaluations of the current landscape in SL and the respective positions of the ‘pro’ and ‘con’ camps. However, the article needs to be revised. Many claims are made without reference to the rich bodies of existing literature and without clearly stating how this data was gathered. In its current form it reads like an evaluative desk-study with some expert interviews.
Additionally, some claims are seriously overstated due to reductive assessments. Generally, the role and voices of women in supporting or opposing Bondo (which is reduced to a discussion of cutting) are always put in the shadow of either male figures (pro) or human rights discourses (con). This misrepresents the situation on the ground. Instead, the author should analyse how, historically, female politicians and wives of presidents championed the Sande, never men on their own account. In contemporary politics it continues to be the wives of presidents who support or initiate campaign while men tend to rephrase what powerful female leaders have said (see former president Koroma's wife). The author further claims that “Since independence politicians have used the strategy of financially supporting the mass initiation of young girls into the Bondo society, just before an election.” This is a false claim. Again, it was the wives of presidents and female political authorities who made this possible. Indeed, men rarely dare to decide over this powerful, female institution which produces politicians, judges, military authorities etc.
Please also revisit the way in which cutting is actually undertaken. It is not always an excision. Indeed, a pioneering study which should be included in the lit review argued that “They show that 31.7% (n = 143) respondents had type Ib (removal of the clitoris with the prepuce); 64.1% (n = 289) had type IIb (partial or total removal of the clitoris and labia minora); and 4.2% (n = 19) had type IIc (partial or total removal of the clitoris, the labia minora, and the labia majora).” (https://www.ncbi.nlm.nih.gov/pmc/articles/PMC3800578/).
We need a thorough introduction to the Bondo or the Sande (maybe through a literature review). Several academics have provided such histories and it would be good if your lit review were to reflect that. As it stands, Bondo’s social, economic and political significance is not sufficiently explored. The author states that the initiations have been reduced to the cutting ritual without going into detail about what they used to be. Their significance for members professional and private trajectories through the Bondo’s networks are not mentioned.
The author claims that “Bondo is embodied in a patriarchal ideology that associates women with the body, and with blood and flesh.” I am afraid this claims is not sustained by either local experts no the vast bodies of existing literature. If this is indeed true, show us how.
Generally, the author tends to make a statement about Bondo and then diminish it by saying that ‘this is due to patriarchal hierarchies.’ However, rich bodies of literature have analysed that Sierra Leone’s gendered landscape—and not least because of the Bondo/Sande, Poro and others—shows gender parallelism not hierarchy. Indeed, the author should visit the concept of gender parallelism for which the Upper Guinea coast is well known. Parallelism differs from oppressive hierarchies.
Another example: “Bondo leaders do play an important role in the exclusively male institution of Poro as the very integral office of the “Mabole” is traditionally occupied by senior Bondo leaders, who are believed to have knowledge of traditional healing herbs needed in the performance of Poro rituals. The reverence of Bondo to the Poro society is articulated in a very popular saying in Sierra Leone- “Nar Bondo born Poro” ( Bondo gave birth to Poro). However, this symbolic power is subordinated when a woman is to become a paramount chief and is expected to give up her Bondo membership to be initiated into the Poro, thereby clearly manifesting that leadership is masculine and male dominated.”
Here again, a key element of parallelism which, if anything, respects women as more capable and indeed powerful, is diminished by using a concept of hierarchy from the geopolitical West to describe a local practice of parallelism. To sustain such claims show us exactly how they can be sustained.
Re- the Mabole. The author writes that “However, this symbolic power is subordinated when a woman is to become a paramount chief and is expected to give up her Bondo membership to be initiated into the Poro, thereby clearly manifesting that leadership is masculine and male dominated.” This is very different than Ferme’s in depth analysis of the Mabole (2001) which is not mentioned at all.
Indeed, many researchers have already explored the questions the author poses. To understand the original contribution, we need a review of this literature and an explanation of how this paper differs. I think it has the ability to offer such a contribution for instance by foregrounding the legal discourse and the discourse on rights and culture more (This would however require including Cowan’s work ‘culture after rights’). Rephrasing the article in this way would be an option to tease out its unique contribution.
The points made around Madame Yoko are interesting, but they are over pressed. Not only did Madam Yoko pair novices with elder men, but she also increased the reach and power of the Bondo society beyond recognition and not just as a tool to gain power but to increase female power and decision making. Many of her novices went on to take up powerful roles within the socio-political sphere across Mende territory.
This article has the potential to make an interesting contribution, but it has to be revised. We need to be much clearer on how the data was collected and with which driving questions. Who was spoken to and why? Further in which parts of SL was the data collected as the divisions regarding FGC tends to differ greatly between locations. The author tends to lump people together into representative groups such as ‘politicians’ ‘activists.’ But who are they and who do they speak for?
We either need a more refined analysis of female authority, bondo-poro relationality and gendered structures (hierarchy versus parallelism). Or these parts should be taken out and the argument rephrased around the issue of rights, culture, choice etc. This is where I see the potential of the article. If this is done with reference to the academic debate around culture and rights (the collection is cited but not sufficiently employed and the recent addition is missing altogether), it could turn out to be a very nice piece.
In any case, a much more detailed and discussion of the literature should be undertaken (where is Ferme [Mabole], Bjälkander, Bosire [political function of the Bondo]……)
I suggest the author R&R.
Round 2
Reviewer 1 Report
The author has made only minor changes to the article. That's
really not enough to address the multiple issues described in the first
round of comments. This article would still need a major revision to be
publishable.
Author Response
I have revised my paper as suggested by the reviewer

Reviewer 2 Report
The author has made changes to the document. I believe with some more substantive editing the paper would have made a more significant contribution but agree that it can be published now.
Author Response
I have revised as advised by the reviewer
